# Immune Modulatory Effects of Molecularly Targeted Therapy and Its Repurposed Usage in Cancer Immunotherapy

**DOI:** 10.3390/pharmaceutics14091768

**Published:** 2022-08-24

**Authors:** Tiancheng Zhang, Chenhao Zhang, Zile Fu, Qiang Gao

**Affiliations:** 1Department of Liver Surgery and Transplantation, Key Laboratory of Carcinogenesis and Cancer Invasion (Ministry of Education), Liver Cancer Institute, Zhongshan Hospital, Fudan University, Shanghai 200032, China; 2Key Laboratory of Medical Epigenetics and Metabolism, Institutes of Biomedical Sciences, Fudan University, Shanghai 200032, China; 3State Key Laboratory of Genetic Engineering, Fudan University, Shanghai 200433, China

**Keywords:** targeted therapy, immunotherapy, drug repurposing, drug combinations, immune checkpoints inhibitor (ICI)

## Abstract

The fast evolution of anti-tumor agents embodies a deeper understanding of cancer pathogenesis. To date, chemotherapy, targeted therapy, and immunotherapy are three pillars of the paradigm for cancer treatment. The success of immune checkpoint inhibitors (ICIs) implies that reinstatement of immunity can efficiently control tumor growth, invasion, and metastasis. However, only a fraction of patients benefit from ICI therapy, which turns the spotlight on developing safe therapeutic strategies to overcome the problem of an unsatisfactory response. Molecular-targeted agents were designed to eliminate cancer cells with oncogenic mutations or transcriptional targets. Intriguingly, accumulating shreds of evidence demonstrate the immunostimulatory or immunosuppressive capacity of targeted agents. By virtue of the high attrition rate and cost of new immunotherapy exploration, drug repurposing may be a promising approach to discovering combination strategies to improve response to immunotherapy. Indeed, many clinical trials investigating the safety and efficacy of the combination of targeted agents and immunotherapy have been completed. Here, we review and discuss the effects of targeted anticancer agents on the tumor immune microenvironment and explore their potential repurposed usage in cancer immunotherapy.

## 1. Introduction

Cancer has been a major leading cause of death worldwide [1]. The novel technology allowed researchers to focus on cancer initiation and progression at levels of cellular and molecular phenotype. Furthermore, the concept of hallmarks of cancer attracted attention to the commonalities that are shared among distinct types of cancer, and many drugs were exploited based on these characteristics. Cancer pharmacological treatments can range from chemotherapy to targeted therapy and immunotherapy. In the 1940s, chemotherapeutic drugs raised hope for patients with advanced or metastatic cancer. However, chemotherapy’s toxicity to normal cells plagued clinical doctors for decades until the approval of targeted anticancer agents [2]. Molecularly targeted therapy is a type of cancer treatment to inhibit cancer progression via small-molecule drugs or antibodies [3]. For instance, the anti-angiogenesis agents have been exploited and approved for treating solid tumors, a milestone in the discovery history of molecularly targeted agents [4].

Targeted agents have become first- or second-line treatments for most advanced malignancies, including breast cancer, lung cancer, colorectal carcinoma (CRC), hepatocellular carcinoma (HCC), renal cell carcinoma (RCC), and others [5]. Targeted agents exert broad anti-tumor activity via direct inhibition on tumor cells and indirect impacts on the tumor microenvironment. Agents approved in clinical use mainly act by inhibiting cycle-dependent kinase (CDK), KRAS, and PI3K signaling, DNA damage repair (DDR) and apoptosis, and ErbB family signaling [6]. In addition, many drugs do not belong to the above four categories but are widely used in clinical treatment, such as the Bruton tyrosine kinase (BTK) inhibitor. During cancer cell invasion and spread, multiple signaling pathways are involved. Indeed, co-targeting different molecules induced synergetic effects, and multi-targeted drugs showed a significant clinical advantage [7]. However, drug resistance caused by targeted agents results in limited response rates and duration of response, especially for patients with advanced or aggressive malignancies. For instance, MET amplification has been proven to be the mechanism of primary resistance to epidermal growth factor receptor (EGFR) tyrosine kinase inhibitor (TKI) therapy in EGFR-mutant non-small cell lung cancer (NSCLC) patients.

Beyond blocking oncogenic mutations or transcriptional targets, molecularly targeted agents modulate the immune context in the tumor microenvironment, also known as the tumor immune microenvironment (TIME). For example, CDK4/6 inhibitors “inflame” TIME by recruiting immune effector cells and suppressing Treg cell proliferation. With a deeper understanding of various subsets of immune cells, there has been a recent surge of interest in exploring the role of TIME in tumorigenesis and metastasis. As an example, to evade the attack of effector immune cells, tumor cells can produce immune suppressive factors, such as IL-6, IL-8, and transforming growth factor-β (TGF-β) to recruit immunosuppressive cells and impair the anti-tumor immune response, favoring tumor progression [8,9]. Thus, immunotherapy is based on the studies of how immunity recognizes and eliminates cancer cells and the mechanism of cancer cells’ evolution to avoid the attack. The identification of immune checkpoints in TIME, to some extent, reveals the mechanism that tumor-specific T cells cannot eliminate cancer cells efficiently without pharmaceutical interventions.

The clinical success of ICIs has brought a revolution in cancer therapy, and it is widely used as a standard treatment in various cancers. Until now, the targets of FDA-approved agents focus on CTLA-4 (ipilimumab), PD-1 (pembrolizumab, nivolumab, cemiplimab, etc.), PD-L1 (atezolizumab, durvalumab, avelumab), and LAG-3 (relatlimab) [10,11]. Despite the great progress in recent years, using ICIs as monotherapy is still limited due to an unsatisfactory response rate. Based on preclinical and clinical data, it is interesting that most targeted anticancer agents could enhance the patient response to ICIs. For example, venetoclax, the first FDA-approved BCL2 inhibitor, increased the T effector memory cells and showed great potential in combination with ICIs [12]. What is more, according to the results of IMbrave150, a phase 3 clinical trial, the FDA approved the combination of atezolizumab, selectively targeting PD-L1, and bevacizumab, a VEGF-A-targeting monoclonal antibody, for first-line treatment in patients with unresectable or metastatic HCC [13]. In contrast, KEYNOTE-240, another phase 3 clinical trial, tested the efficacy and safety of pembrolizumab, an anti-PD-1 monoclonal antibody, as monotherapy for advanced HCC, and its result did not reach statistical significance [14].

The existing molecularly targeted agents are designed to block the signaling pathways involved in hallmarks of cancer, such as inducing angiogenesis, immune evasion, and metabolic reprogramming. It is not surprising that the repurposing of targeted therapy combined with immunotherapy may become the future paradigm and direction of clinical investigation. In this review, we summarize how the targeted anticancer agents influence the tumor microenvironment and the potential combination of targeted therapy with immunotherapy.

## 2. Modulatory Effects of Targeted Therapy on Immune Cells

TIME plays a vital role in the modulation of tumor initiation, progression, and drug resistance. The generation of immune response to cancer is a multistep process and the last step is to eliminate cancer cells, which occurs in TIME and is an important rate-limiting step. Using ICIs released the restriction of effector T cells, and thereby many cancer patients benefit from this approach [10,15]. However, a few cancer types, such as pancreatic ductal adenocarcinoma (PDAC), neuroendocrine neoplasm, and mismatch repair-proficient CRC, barely benefit from immunotherapy, which may be due to the “cold” immune phenotype that contains fewer effector immune cells and more immunosuppressive cells [10,16]. It is possible to reuse targeted drugs to inhibit the specific type of cells, molecules, or pathways in TIME, enhancing the efficacy of immunotherapy. An in-depth investigation of targeted anticancer agents’ immunosuppressive and immunostimulatory capacity may provide the theoretical basis for the combinatorial treatment with immunotherapy (Figure 1).

### 2.1. Effector T Cells

Cytotoxic CD8+ T cell is the key to adaptive immunity against tumor cells. Notably, eradicating tumor cells by effector T cells is a multistep process; many immunosuppressive factors directly or indirectly counteract this process. For example, activated T cells in TIME obtain energy through glycolysis even in the presence of oxygen after metabolic reprogramming, and the block of T cells glycolysis attenuates their anti-tumor activities [17]. A subset of T cells in a state of exhaustion is defined as the loss of effector function, such as cytotoxicity, and correlates with the resistance to immunotherapy. Consistent with these notions, improving the infiltration of tumor-specific T cells, maintaining activated T cells metabolism, reversing effector T cells exhaustion, and activating T cells’ tumor-killing capacity have long been the purposes of developing new therapeutic strategies.

To date, three CDK4/6 inhibitors, including palbociclib, ribociclib, and abemaciclib, have been approved by FDA. Clinical trials demonstrate that CDK4/6 inhibitors prolong the progression-free survival in patients with estrogen receptor (ER)-positive breast cancer when combined with anti-estrogen therapy. The immunostimulatory activity of CDK4/6 inhibition can be achieved by interacting with T cells via the depression of the nuclear factor of the activated T cell (NFATC) family [18]. In the TIME of tumors treated with the BRAF inhibitor, anti-tumor-specific CD8+ T cells were enriched due to increased tumor antigen expression [19,20]. Beyond using BRAF inhibitors singly, the BRAF(V600E/K)-mutant melanoma patients benefited from the combinational treatments of BRAF inhibitor and MEK inhibitor [21]. It has been proved that the immunoregulatory effects induced by combinatorial therapy are vital to tumor shrinkage [22]. This notion is supported by the combination of BRAF and MEK inhibition, which boosts the expansion of tumor-specific T cells and the abundance of TCR repertoire, which is due to the induction of TCF7 and T-bet [23,24]. Pharmacological inhibition of MDM2 by ALRN-6924 improves the infiltration of cytotoxic CD8+ T cells, reflecting the potential for MDM2 inhibitors combined with immunotherapy [25]. The existing evidence suggests that VEGF/VEGFR signaling decreases the proliferation of T cells and increases the expression of PD-1 on CD8+ T cells [26]. In both lung and CRC murine models, BD0801, an anti-VEGF mAb, showed a significant anti-tumor capacity when combined with ICIs, whose potential mechanism is increasing CD8+ effector T cells and reducing CD8+PD-1+ exhaustion T cells [27]. Sorafenib and Lenvatinib, two multi-kinase inhibitors, are the first-line drugs for advanced HCC [28]. In the mouse HCC model, sorafenib showed the ability to deplete PD-1+CD8+ T cells, enhancing the anti-tumor immune response [29]. Indeed, the clinical data showed that the number of CD8+Ki67+IFN-γ+ T cells increased after treatment with sorafenib, and the upregulation of CD8+Ki67+IFN-γ+ T cells was associated with better patient outcomes [30]. Lenvatinib has been proven to augment anti-tumor activities by increasing T cell infiltration and decreasing the number and immunosuppression of myeloid-derived suppressor cells [31]. Cancers harboring epidermal growth factor receptor (EGFR) mutation always build a non-inflamed TIME [32]. The treatment with EGFR inhibitor remodels the TIME by improving the numbers of T cells and decreasing the immunosuppressive cells [33]. Conversely, in the breast cancer mouse model, the combination of PARP1 and PARP2 resulted in remolding the TIME and less activation of T cells [34]. Altogether, these findings suggest that targeted drugs could modulate the effector T cells priming, differentiation, and functions. Notably, among quantities of targeted molecules, the agents targeting ErbB augment T cell-based anti-tumor activities from multiple aspects, such as increasing the number of CD8+ T cells, enhancing cytotoxic T cell function, and reducing T cell exhaustion, especially in NSCLCs and CRCs [35].

### 2.2. Regulatory T Cells

Regulatory T cells (Treg) control autoimmunity and suppress the inflammation response, which maintains the hosts’ immune homeostasis. However, in TIME, Treg cells cooperate with other immunosuppressive cells to promote tumor growth and metastasis. It has been proven that chemotherapy such as cyclophosphamide (CTX) can obliterate Treg cells and augment anti-tumor activities [36]. The low dose of CTX attenuated the suppression of immunity induced by CD4+CD25+ Treg cells. It promoted the aggregation of activated T cells, which has been used as the pretreatment of a cancer vaccine [37,38]. The mechanism of immune invasion that progressive cancer recruits Treg cells could be understood as a host’s “self-identity” to tumor cells. A deeper insight into the biology of Treg cells reveals that using agents targeting the critical process of Treg cell function is a promising strategy.

The effect of CDK4/6 inhibitors on Treg cells has been tested both in vitro and in vivo. The underlying mechanism for the depletion of Treg cells is the overexpression of p21 and repression of DNA methyltransferase 1 (DNMT1) [39]. From the analysis of RIBECCA trial, ribociclib, a CDK4/6 inhibitor, significantly downregulated peripheral CD4+FOXP3+CD25+ Treg cells [40]. Beyond increasing T cell infiltration, another potential mechanism of lenvatinib-mediated immunoenhancement is reducing tumor intrinsic Treg cells, which opens up an opportunity to combine it with immunotherapy [41]. Moreover, the selective-PI3K inhibitor ZSTK474, in an optimal dose, deletes Treg cells specifically, and, intriguingly, PI3K inhibition promoted the differentiation into tumor-specific CD8+ memory T cells, which contributed to a more robust immune response [42]. A recent clinical study showed that intermittent use of AMG319, a novel PI3Kδ inhibitor, may result in anti-tumor immunity by reducing intratumoral Treg cells with controllable adverse events [43]. The evidence showed that aberrant EGFR signaling increased the frequency of Treg cells in TIME by chemokine production, therefore EGFR inhibitors suppressed this process, improving the response of ICIs in NSCLC patients [35]. Altogether, the evidence that Treg cells support tumor progression and restraining Treg cells or their function leads to tumor regress argues that these molecules or pathways could be promising targets. However, it is important to note that the high frequency of Treg cells correlated with a good prognosis in ER- breast cancer and mismatch repair-proficient CRC [44,45,46].

### 2.3. B Cells

Depending on the composition of the tumor microenvironment, tumor-infiltrating B cells (TIBs) play the opposite role of suppressing or promoting cancer cells. TIBs can exert anti-tumor ability by producing the tumor-specific antibody, assisting T cell priming and activation, and killing cancer cells directly [47]. B cells are critical to the formation of the tumor-associated tertiary lymphoid structure (TLS), which supports the anti-tumor activity and may suggest better patient outcomes [48]. IL-10 is considered an immunosuppression cytokine, and the subset of B cells that produce IL-10 is recognized as regulatory B cells (Breg). Unlike Foxp3, the marker of Treg cells, the definitive marker of Breg cells is still lacking. Bregs can induce an immune-suppressive milieu in TIME by inhibiting cytotoxic T cell or cytokine secretion.

Ibrutinib is a covalent and irreversible BTK inhibitor and has been approved to treat mantel cell lymphoma and chronic lymphatic leukemia [49]. Ibrutinib breaks the cross-talk between B cells and FcRγ+ macrophages, activating T cell-based antitumor response in PDAC mouse models [50]. Targeting CDK4/6 enhanced antitumor immunity by inducing a pro-inflammatory environment, where the recruitment of B cells was impressive [51]. This is attributed to CXCL13 secreted by abemaciclib-treated mouse ovarian cancer cells, which support B cell homing to follicles and subsequent T cell activation [51]. Studies on the effects of targeted anticancer agents on TIBs are insufficient. Existing evidence emphasizes the importance of TLS in response to immunotherapy [52,53]. Correspondingly, understanding how anticancer agents influence B cell infiltration and formation of TLS is crucial for further studies.

### 2.4. Natural Killer (NK) Cells

Natural killer (NK) cells play an essential role in innate immunity, which acts as the “killer” to control tumor growth and metastasis. Based on their surface expression level of CD56, NK cells are usually divided into two subsets, including CD56^dim^ NK cells and CD56^bright^ NK cells [54,55]. NK cells participate in innate and adaptive immune responses, and their anti-tumor activities do not require antigen-sensitization and are not limited by the major histocompatibility complex (MHC). In addition, the chimeric antigen receptor (CAR) technology used in NK cells shows more advantages than in T cells, such as a broader range of sources of cells and a lower incidence of therapy toxicity [56]. Targeting NK cells could therefore generate a new promising approach to augmenting anti-tumor treatment.

In murine models of KRAS-mutant lung cancer, combined MEK and CDK4/6 inhibition trigger innate immune response, especially by NK cells [57]. Interestingly, retinoblastoma (RB)-mediated cellular senescence contributed to NK cell-mediated cytotoxicity, and this study indicates that specific targeted anticancer agents could remodel the disabled immune surveillance [57]. While there are no selective inhibitors to CDK8, NK cell-specific CDK8 deletion promotes the secretion of perforin in vitro. It improves the outcomes in models of melanoma, lymphoma, and leukemia [58]. Tumor necrosis factor (TNF)-related apoptosis-inducing ligand (TRAIL) could upregulate the percentage of NK cells in TIME when used in combination with CDK inhibitor in NSCLC mouse models [59]. Supporting the efficacy of the BRAF inhibitor against BRAF(V600E)-mutant melanoma, a recent study suggested that BRAF inhibition directly enhanced NK cells’ immunostimulatory capacity via upregulating CD69 expression and IFNR release [60]. Ovarian cancer cells pre-treated by EGFR TKIs are more sensitive to NK cell-mediated antibody-dependent cell-mediated cytotoxicity (ADCC), which points to an exciting possibility of designing new combined immunotherapy [61]. Conversely, the expression of an NK-cell-activating ligand, MICA, and DNAM-1, was downregulated after the treatment of vemurafenib [62]. PI3K∂-selective inhibitor idelalisib attenuated the proliferation and cytotoxicity of NK and T cells, potentially explaining the limited efficacy of idelalisib [63]. The histone deacetylase inhibitor (HDACi) has been used to treat several hematological malignancies. However, HDACi upregulated NKG2DLs on tumor cells in vitro and impaired NK cells’ viability in vivo [64]. Further works about targeted drugs’ exact effects on NK cells can accelerate the rational development of NK cell-based immunotherapy.

### 2.5. Neutrophils

Neutrophils, as fast-response immune cells, obliterate different kinds of pathogens but may play both pro-and anti-tumor roles in the TIME. By virtue of different functions, neutrophils are classified using several terms, including N1/N2 neutrophils, tumor-associated neutrophils (TANs), and polymorphonuclear neutrophil myeloid-derived suppressor cells (PMN-MDSCs) [65].

Because of the side effects in normal organs, the use of TGF-β inhibitors is limited. Interestingly, neutrophils are utilized as vehicles to load TGF-β inhibitors and then transport drugs to tumor sites [66]. Nanomedicine that can locally release TGF-β was developed and showed potential in boosting neutrophils from the N2 to N1 phenotype in vitro and in vivo [67]. Conversely, a high dose of VEGFR2 inhibitor, apatinib, induced the expression of IL17A by γδ T cells, which resulted in neutrophil polarization to N2 phenotype and the exhaustion of CD8+ T cells in breast cancer mouse models. VEGFR inhibitors boosted the migration of neutrophils, and neutrophils could establish the pre-metastasis niche in favor of circulating tumor cells to seed [68].

Recent studies highlight that driving reverse migration is a more reasonable strategy than the depletion of neutrophils and preventing neutrophil infiltration [69,70]. Regarding the plasticity of neutrophils in TIME, finding specific drugs that induce the transformation of N2 to N1 TANs is expected to be another direction. Single-cell analysis of PDAC patients’ samples demonstrates that pro-tumor TANs are prone to exhibiting high glycolytic activity. Cancer cells can induce the glycolytic switch of TANs to mediate immunosuppressive TIME, which may open up a new mode of therapy. Altogether, with a much clearer understanding of TAN biology, we may anticipate benefits from molecularly targeted drugs that modulate TANs.

### 2.6. Dendritic Cells

Dendritic cells (DCs) function in antigen recognition and presentation and initiate an adaptive immune response [68]. While cancer cells present antigens onto their major histocompatibility complex I (MHCI), it is vital that DCs cross-present the tumor antigens to prime CD8+ T cells. Taking into account the latest therapy strategies, DC vaccines show potent tumor growth inhibition [71].

Trastuzumab, targeted to the extracellular domain of HER2, mediated immunostimulatory based on recognition and cross-presentation of tumor antigen by DCs, which was especially realized through Fc receptor-mediated mechanism [72]. Exiting evidence implies that various PI3K isoforms regulate DCs’ functions [73]. A pan-PI3K inhibitor, copanisib, dramatically upregulated CD80, CD86, MHC-I, and MHC-II in DCs, and its combination with ICIs showed a potent antitumor response in bladder cancer mouse models [74]. A combination of BRAF and MEK inhibition promoted the maturation of DCs and T cell activation, mainly reflecting the potential of their combination with immunotherapy [24]. Currently, plenty of DC-based immunotherapies have been developed and tested in clinical trials. It is a great supplement to discovering targeted agents that modulate DCs’ anti-tumor capacity to improve DC maturation and antigen presentation.

### 2.7. Macrophages

Traditionally, macrophages are roughly divided into antitumor (M1) or protumor (M2) categories. With the development of single-cell technology, it has been realized that dynamic changes in macrophage phenotypes occur during tumorigenesis and progression, and different TAMs subpopulations are responsible for distinct functions [75]. The molecularly targeted agents that inhibit or alert TAMs pro-tumoral activities need extensive investigation.

Blocking the recruitment of macrophages is an efficient approach to deplete TAMs. VEGF contributes to the attraction of monocytes by VEGFR-1 and promotes angiogenesis. Surprisingly, anti-VEGF therapy induced macrophages to move to the tumor via the establishment of tumor hypoxia [76]. Several studies have shown solid evidence that repolarizing macrophage leads to a promising direction to augment immunotherapy. In the murine CRC model, CD11b+ Ly6C+ MHC-II+ cells, an intermediate state from monocytes to TAMs, accumulated in the tumor environment, where MEK inhibition suppressed macrophage polarization [77]. Interestingly, CD11b+ FcγRI/III+ myeloid cells were discovered with increasing PI3Kγ signaling in PDAC [50], where PI3K inhibitor reprogramed macrophages and improved the response to chemotherapy [78]. Given the crucial role of P53 activation in macrophages, the MDM2 inhibitor reprogramed M1-like macrophages into M2-like macrophages in syngeneic mouse models of CRC and HCC [79].

Conversely, using EGFR targeting monoclonal antibodies may activate M2 macrophages, which promote tumor progression [80]. In the TIME of BRCA-associated triple-negative breast cancer (TNBC), a macrophage is a major population of immune cells. Multi-omics data showed that PARP inhibitors alter the anti-tumor activities of macrophages through metabolic reprogramming, which shapes an immunosuppressive environment, but provides a potential therapeutic strategy that targets TAMs in BRCA-associated TNBC [81]. In the future, investigating the dynamic polarization of TAMs and their functional phenotypes could significantly accelerate the discovery of novel therapeutic strategies.

### 2.8. Myeloid-Derived Suppressor Cells (MDSCs)

MDSCs exert all their skills to promote tumor growth and metastasis, such as dampening immune response and facilitating tumor angiogenesis, raising great interest in developing a therapeutic strategy that targets MDSCs. Studies have elucidated that MDSCs are observed in lymphoid tissues, bone marrow, and peripheral blood, in addition to TIME. MDSCs are classified into three categories: Monocytic-MDSCs (M-MDSCs), granulocytic-MDSCs (G-MDSCs), and early immature-MDSCs (eMDSCs) [82]. There is another classification based on morphological and phenotypical traits, and MDSCs are recognized as polymorphonuclear MDSC (PMH-MDSC) and monocytic MDSC (M-MDSC) [82].

Combined CDK 4/6 and PI3K inhibitors significantly depleted MDSCs in TNBC [83]. In a KRAS-driven murine lung cancer model, trametinib, a MEK inhibitor, directly depleted MDSCs [84]. The potential mechanism includes trametinib acting on the MEK pathway, which is necessary for the expansion of MDSCs, and trametinib’s influences on the secretion of osteopontin leading to MDSC depletion [84]. Combined BRAF, MEK, and CDK 4/6 inhibition showed promising anti-tumor activity, driven by reducing the population of tumor-intrinsic myeloid cells in syngeneic mouse models of melanomas [85]. A recent study indicated that the portion of G-MDSCs in peripheral blood was decreased with the treatment of bevacizumab-containing regimens compared with non-bevacizumab-based regimens in patients with NSCLC [86]. Targeted anticancer agents not only affect the number of MDSCs but also their immunosuppressive capacity. Treating mice with a VEGFR inhibitor, axitinib, reprogramed MDSCs toward an antigen-presenting phenotype [87]. Indeed, the approach to regulating myeloid cell differentiation to eliminate the immunosuppression phenotype is worth exploring.

Despite the immunostimulatory results, targeted drugs interfered with the anticancer response of immune cells. This may be one of the mechanisms that result in positive or negative outcomes in clinical research on combination treatment. However, both the immunostimulatory and immunosuppressive effects of targeted agents are not well understood. Further studies on how molecularly targeted therapy influences TIME are needed.

## 3. Modulatory Effects of Targeted Therapy on Immune Checkpoints

Different drugs have diverse impacts on the expression of immune checkpoints such as PD-1/PD-L1. A CDK4/6 inhibitor, Palbociclib, remodeled the TIME not only by its inhibition of RB phosphorylation but also by upregulating the PD-L1 protein level on tumor cells [88]. Consistent with this result, combined Palbociclib with ICIs regressed tumor growth and prolonged survival in mouse models of prostate cancer [88]. In addition, the PARP inhibitor inactivated GSK3β, which attenuated the proteasome degradation of PD-L1 in vitro and in vivo [89,90]. PARP inhibitors, Olaparib and Rucaparib, have been proven to upregulate PD-L1 expression in small and non-small cell lung cancer [91,92]. Another study also indicated that a DNA double-strand break due to treatment with PARPi was involved in the increased PD-L1 expression [93]. The evidence suggested that dabrafenib and trametinib, alone or in combination, were promising treatments for metastatic melanoma [94]. The upregulation of PD-L1 on melanoma cells was due to sustained exposure to dabrafenib, trametinib, or dabrafenib plus trametinib, and the increased level of PD-L1 may be a biomarker of acquired resistance to these drugs [95]. Conversely, cetuximab and erlotinib, EGFR inhibitors, blocked the tyrosine phosphorylation and suppressed the MAPK signaling, which led to the degradation of PD-L1 mRNA [96]. MEK1/2 inhibitors prevented IFNγ-induced PD-L1 mRNA upregulation, whose mechanism differs from the EGF-induced PD-L1degradation [96]. PRM2 promoted the expression of PD-L1 via the RRM2/ANXA1/AKT axis [97]. Though there is no evidence that the existing PRM2 inhibitors function effectively in anti-tumor activities [98,99], it offers a direction to develop a new PRM2 inhibitor. c-MET promoted the progression of HCC and the high expression of MET correlated with short patient survival [100,101]. However, a phase 3 study of the nonselective MET inhibitor tivantinib in patients with MET-high and previously treated advanced HCC showed that tivantinib did not improve overall survival [102]. A recent study demonstrates that MET showed a negative correlation with PD-L1 in HCC and prolonged the survival in HCC mouse models by simultaneously blocking MET and the PD-1/PD-L1 pathway [103].

Treatment with the MEK inhibitor in a pulsatile way showed better control of tumor proliferation than continuous treatment [104]. CTLA-4 expression was upregulated after the abovementioned treatment in vitro and in vivo [104]. Decitabine, a DNMT inhibitor, induced the up-regulation of CTLA-4 in a dose-dependent way [105]. Correspondingly, Decitabine combined with the CTLA-4 blockade enhanced the innate and adaptive immune response in murine ovarian cancer models [106]. Taken together, immune checkpoints are regulated by various molecularly targeted drugs in different cancer types. Thus, molecularly targeted agents’ cancer-specific immunomodulatory role needs further investigation, especially combined with immunotherapy.

## 4. Potential Combinations of Targeted Therapy and Immunotherapy

In the past decade, significant progress in developing targeted drugs and ICIs has enhanced the diversity of treatment regimens. Recent clinical trials have demonstrated that long-term tumor remission with tolerable side effects can be achieved by combining targeted therapy and immunotherapy.

### 4.1. CDK Inhibitors with ICIs

CDKs are a family of protein kinases involved in cell cycles, gene transcription, insulin secretion, glycogen synthesis, and neuronal functions, including 21 CDKs and 5 CDK-like genes [107]. The overexpression or deregulation of CDKs is generally considered to lead to unlimited proliferation and dysregulated bioactivity in several tumor types, such as breast cancer, CRC, and sarcoma [108,109]. Several CDK inhibitors were developed, among which the CDK4/6 inhibitors palbociclib, ribociclib, and abemaciclib were recommended as vital components of systemic therapy for HR+ breast cancer. In a recent phase I/II clinical trial (NCT02778685), combination therapy of palbociclib, pembrolizumab, and Letrozole showed good tolerance and a good objective response rate (ORR = 56%) as the first-line treatment for HR-positive metastatic breast cancer [110]. A phase Ib clinical trial (NCT02779751) evaluated the combination of abemaciclib and pembrolizumab in patients with NSCLC and HR+ and HER2− breast cancer. Results showed that abemaciclib plus pembrolizumab achieved a satisfactory ORR of 14.3% with a manageable safety profile in HR+ and HER2− breast cancer patients [111]. At the same time, the combination led to greater toxicity than each agent alone in NSCLC patients [112]. In summary, the contrary results in breast cancer and NSCLC patients demonstrated both anti-tumor activity and the possible toxicity of combination treatment, which requires more risk–benefit consideration (Table 1). To conclude, the combination of ICIs and CDK inhibitors may soon be applied in clinical use for breast cancer. More evidence is needed for lung cancer.

### 4.2. KRAS Signaling Inhibitors with ICIs

RAS is one of the most frequent mutations in human cancer and KRAS is the most frequently mutated isoforms, constituting approximately 86% of all RAS mutations [113]. Oncogenic KRAS mutation activates a series of downstream proteins, including PI3K, AKT, RAF, mTOR, and MAPK. Despite the strong oncogenic effect of KRAS in human cancer, KRAS mutants, except G12C, remain undruggable. Therefore, agents that target downstream signaling pathways are considered possible alternatives.

Mutated BRAF inhibitors dabrafenib and vemurafenib were approved for treating BRAF(V600E) metastatic melanoma [114,115]. Dabrafenib also showed clinical activity in NSCLC patients with BRAF(V600E) mutation [116]. Trametinib, a MEK1/2 inhibitor, showed antitumor activity against BRAF(V600E)-positive cancer [117]. Dabrafenib plus trametinib is now applied as one of the first-line therapies for advanced NSCLC or melanoma with a BRAF(V600E) mutation [118,119]. Intriguingly, a recent report illustrated that the activation of MAPK signaling is a compensatory mechanism of PI3K-Akt inhibition, leading to drug resistance to dabrafenib plus trametinib, and the co-administration of two pathways’ inhibitors shows synergistic anti-tumor activity [120,121]. A recent phase III clinical trial (NCT02224781) reported that the treatment of nivolumab plus ipilimumab followed by dabrafenib plus trametinib resulted in a higher 2-year OS (Overall Survival) rate (72% vs. 52%, respectively; *p* = 0.0095) than the treatment of dabrafenib plus trametinib followed by nivolumab and ipilimumab for patients with advanced BRAF(V600)-mutant melanoma [122]. A phase II clinical trial (NCT02130466) in BRAF-mutant melanoma showed that the triplet combination of dabrafenib, trametinib, and pembrolizumab had a longer median PFS (Progression-free Survival) (16.0 vs. 12.5 months, respectively; *p* = 0.043) and a higher response rate (59.8% vs. 27.8%, respectively) with a higher rate of grade 3/4 adverse events (58.3% vs. 26.7%, respectively) than the doublet combination of dabrafenib and trametinib [123]. A longer median PFS (16.2 vs. 12.0 months, respectively; *p* = 0.042) with a higher rate of grade 3 adverse events (55% vs. 33%) were also reported in dabrafenib, and trametinib plus spartalizumab than dabrafenib and trametinib for BRAF(V600)-mutant advance melanoma [123]. These results emphasized that adverse events could prevent the wide application of KRAS signaling inhibitors with ICIs, but the combination might be an alternative therapy under certain circumstances.

Cobimetinib, another MEK inhibitor, was approved for the treatment of metastatic melanoma in combination with vemurafenib based on a phase III clinical trial [124]. However, cobimetinib plus atezolizumab did not demonstrate a longer PFS than pembrolizumab alone in BRAF wild-type melanoma (5.5 vs. 5.7 months, respectively; *p* = 0.30), which showed that the combination of a MEK inhibitor and ICIs might not be more effective in these patients [125]. The phase III IMspire150 clinical trial (NCT02908672) investigated the possibility of the combination of cobimetinib, vemurafenib, and atezolizumab for advanced melanoma, and the median PFS was significantly prolonged in the triplet group than the conventional combination of cobimetinib and vemurafenib (15.1 vs. 10.6 months, respectively; *p* = 0.025) [126]. In biliary tract cancers, cobimetinib plus atezolizumab improved the median PFS (3.65 vs. 1.87 months, respectively; *p* = 0.027) compared with atezolizumab monotherapy based on a phase II trial (NCT03201458) [127]. A phase II study of cobimetinib plus atezolizumab for NSCLC patients is still ongoing (NCT03600701) [128]. Several other KRAS signaling inhibitors were approved for clinical use, including a PI3K inhibitor, alpelisib, and an mTOR inhibitor, everolimus. However, the safety and efficacy of these targeted agents in combination with ICIs remain unclear.

### 4.3. ErbB Family Inhibitors with ICIs

Overactivation of the ErbB protein family drives the tumorigenesis and development of various malignancies, including breast cancer, CRC, head and neck squamous cell carcinoma (HNSCC), and NSCLC [129]. The ErbB protein family consists of four receptor tyrosine kinase members: ErbB1/epidermal growth factor receptor (EGFR), ErbB2/HER2, ErbB3, and ErbB4. Over the past two decades, TKIs and monoclonal antibodies targeting EGFR and HER2 were developed and applied for the front- and subsequent-line treatments of human cancers.

Approved EGFR inhibitors include gefitinib, erlotinib, osimertinib, lapatinib, vandetanib, cetuximab, panitumumab, and necitumumab [130]. A phase I study (NCT02088112) reported that median PFS was 10.1 months, and ORR was 63.3% in NSCLC patients receiving gefitinib plus durvalumab [131]. No significant increase in PFS with higher toxicity was reported in the combination of gefitinib plus durvalumab. A phase I study (NCT02040064) reported that gefitinib plus tremelimumab limitedly improved survival (median PFS = 2.2 months) with a high adverse event rate [132]. In a phase I/II KEYNOTE-021 clinical trial (NCT02039674), gefitinib plus pembrolizumab was not feasible in NSCLC patients because 71.4% of patients had grade 3/4 liver toxicity [133]. Moreover, erlotinib plus pembrolizumab did not improve drug response compared with monotherapies. These results showed that a combination of gefitinib and ICIs might not be practicable due to side effects and that the combination of erlotinib and ICIs needed more evaluation. A retrospective study reported that osimertinib after nivolumab increased the frequency of hepatotoxicity in NSCLC patients [134]. A phase II clinical trial (NCT03082534) reported that cetuximab plus pembrolizumab showed promising activity for advanced HNSCC patients [135]. A phase Ib/II study (NCT02713373) showed that cetuximab plus pembrolizumab was well tolerated in CRC patients [136]. In a phase II trial (NCT03442569), the combination regimen of panitumumab, ipilimumab, and nivolumab showed a good median PFS (5.7 months) [137]. In a phase II study in advanced NSCLC patients, necitumumab and pembrolizumab showed potential benefits (median PFS = 4.1 months) with no additive side effects as second-line therapy [138]. According to the present evidence, ICIs combined with EGFR TKIs may cause more adverse effects, while ICIs and EGFR mAbs are promising treatments with favorable benefits and controllable toxicity (Table 1).

Approved HER2 inhibitors include neratinib, tucatinib, afatinib, pertuzumab, and trastuzumab, which target the overexpression of HER2 receptors in NSCLC, breast cancer, and gastric cancer [139]. In the phase II ALPHA clinical trial (NCT03695510), afatinib plus pembrolizumab demonstrated potential benefits for advanced HNSCC patients, with 53.8% of patients showing an objective response [140]. In the phase Ib/II PANACEA trial (NCT02129556), the addition of pembrolizumab brought durable clinical benefit to trastuzumab-resistant HER2+ breast cancer patients [141]. Pembrolizumab plus trastuzumab also showed a similar trend as the first-line treatment of esophageal, gastric, or gastro-esophageal junction cancer (NCT02954536) [142]. However, these results did not include control groups, and further trials are required to assess the efficacy and safety compared with current therapies (Table 2).

### 4.4. PARP Inhibitors with ICIs

PARP plays a vital role in DNA repair pathways, and tumors with defective homologous recombination, especially BRCA-deficient, are susceptible to PARP inhibitors, including olaparib, talazoparib, niraparib, and rucaparib [143]. In a phase I/II clinical trial for recurrent platinum-resistant ovarian cancer (NCT02657889), niraparib plus pembrolizumab showed promising antitumor activity with an ORR of 18% [144]. A phase II clinical trial that investigated the combination of olaparib and durvalumab in advanced prostate and ovarian cancer reported modest efficacy with an acceptable safety profile [145,146]. Several other phase II studies are still ongoing [147,148]. Although there is a long way to go before clinical use, the combination of PARP inhibitors and ICIs may be of great value for patients with specific mutations.

### 4.5. FGFR/PDGFR/VEGFR Signaling Inhibitors with ICIs

The activation of receptor tyrosine kinases, such as FGFR, PDGFR, and VEGFR, is involved in the development and metastasis of various human cancers [149]. Inhibitors of these kinases have been widely applied in treating HCC, RCC, and differentiated thyroid cancer (DTC). A phase Ib clinical trial (NCT03628521) showed the encouraging efficacy (median PFS = 15 months; ORR = 72.7%) of anlotinib plus sintilimab, an anti-PD-1 antibody, with tolerable adverse events in NSCLC patients [150]. Recent phase II trials also reported the efficacy and safety of apatinib plus camrelizumab, an anti-PD-1 antibody, in advanced TNBC (NCT03394287) and HCC (NCT02942329) [151,152]. However, apatinib plus camrelizumab failed to improve PFS and ORR in metastatic CRC in a phase II trial (NCT03912857) [153]. In the phase III KEYNOTE-426 trial (NCT02853331), axitinib plus pembrolizumab showed longer PFS (15.4 vs. 11.1 months, respectively; *p* < 0.0001) than sunitinib monotherapy as a first-line treatment for advanced RCC [154]. Cabozantinib plus nivolumab showed longer PFS (16.6 vs. 8.3 months, respectively; *p* < 0.001) and higher ORR (55.7% vs. 27.1%, respectively; *p* < 0.001) than sunitinib monotherapy as a first-line treatment for advance RCC [155]. Lenvatinib combined with pembrolizumab showed encouraging clinical benefits in PFS, OS, and ORR with manageable toxicity in advanced papillary thyroid carcinoma (PTC), HCC, endometrial cancer, RCC, and gastric cancer according to recent clinical trials [156,157,158,159,160]. In a phase II study, metastatic CRC patients receiving regorafenib plus avelumab showed a median OS of 10.8 months and a median PFS of 3.6 months without unexpected side effects [161]. In a retrospective study, regorafenib plus sintilimab showed better OS (13.4 vs. 9.9 months; *p* = 0.023), longer PFS (5.6 vs. 4.0 months; *p* = 0.045), and higher ORR (36.2% vs. 16.4%; *p* = 0.045) than regorafenib alone as a second-line treatment for advance HCC [162]. Pemigatnib, a selective FGFR inhibitor, showed tolerable toxicity and potential antitumor activity in combination with pembrolizumab for advanced cancer based on the phase I/II FIGHT-101 trial (NCT02393248) [163]. Retrospective studies also revealed that TKIs plus anti-PD-1 antibodies, including nivolumab, pembrolizumab, or sintilimab, showed potential benefits with tolerable toxicity in advanced HCC patients [164]. These results suggested the encouraging efficacy and safety of multi-targeted TKIs combined with ICIs (Table 2).

Bevacizumab works as an antitumor agent in clinical via its blockade of VEGF. Bevacizumab, together with atezolizumab, is recommended as first-line therapy for advanced HCC because the combination showed better OS (64.2% vs. 57.6% at 12 months) and PFS (6.8 vs. 4.3 months; *p* < 0.001) than sorafenib in the phase III trial (NCT03434379) [165]. This combination also improved PFS (11.2 vs. 7.7 months) than sunitinib for advanced RCC patients in the phase III clinical trial (NCT02420821) [166]. Sintilimab, another VEGF mAb, showed improved PFS (6.9 vs. 4.3 months) combined with sintilimab and chemotherapy than chemotherapy alone for NSCLC patients in the phase III study. These results indicated the potential efficacy with good tolerance of VEGF mAb plus ICIs as first-line therapies (Table 2). To conclude, the application of FGFR/PDGFR/VEGFR signaling inhibitors plus ICIs may provide promising benefits with similar adverse events for patients with malignancies.

### 4.6. Epigenetic Agents with ICIs

Epigenetic agents are emerging combination partners for the treatment of malignancies, and recent studies investigated their role in combination with ICIs (Table 3). Entinostat, an HDAC inhibitor, demonstrated promising clinical benefits with an ORR of 9.2% in NSCLC and 14% in metastatic uveal melanoma in combination with pembrolizumab in phase II clinical trials [167,168]. Azacitidine, a DNMT inhibitor approved for acute myeloid leukemia (AML), showed encouraging clinical activity in combination with nivolumab for AML patients but limited efficacy with avelumab (ORR = 10.5%) [169,170]. Decitabine, another DNMT inhibitor, achieved bot ah high response rate (ORR = 52%) and long-term survival improvement (median PFS = 20.0 months) with camrelizumab [169]. These results indicated that anti-tumor activity with acceptable toxicity might be possible with epigenetic agents plus ICIs, especially anti-PD-1 antibodies.

## 5. Discussion

In addition to blocking cell growth, targeted therapy contributes to the TIME remodeling that enhances the anti-tumor response. Treatment response to ICIs is mainly dependent on an active TIME. Therefore, targeted therapy can be a potent option to improve the efficacy of ICIs, possibly through mechanisms including the reinforcement of effector T cell infiltration and the impairment of immunosuppressive cells. Indeed, the strategy is being investigated in many clinical trials, but most of them are in an early stage (Table 1, Table 2 and Table 3). To optimize the combinational therapeutic approach, several issues are noteworthy.

The first is how the proteins or signaling pathways interfered with by targeted agents influence each immune component in TIME and their integrated effect on the antitumor response. Corresponding studies have been conducted in preclinical models and patient cohorts. As an example, VEGF signaling inhibitors, demonstrated to activate effector T cells, promote DC maturation, and boost Treg cell depletion, were approved for treating various cancers in combination with ICIs [27,171]. However, some agents modulate an immunosuppressive TIME by impairing effector immune cells’ function, enhancing the recruitment of immunosuppressive cells and promoting polarization to pro-tumor phenotype [34,80].

The second is whether the increasing anti-tumor activities of the combination strategy escalate the toxicities. The side effects induced by molecularly targeted agents are attributed to the inhibition of targets in normal tissues, such as rash, hypertension, and hepatoxicity. Immune checkpoints protect the body from damage as a consequence of dysregulated immunity. Therefore, the immune-related adverse events (irAEs) coupled with ICIs may be due to an imbalance in immunologic homeostasis. Assessing the treatment safety and finding biomarkers of side effects is important.

Selecting appropriate patient subgroups for combined treatment and discovering potential biomarkers to predict the safety and efficacy are essential. Without synergy effects of the drug combination, choosing patient subgroups precisely can also contribute to the improved response of combination therapy [172]. Thus, beyond molecular features of cancers, more elements that were previously ignored are being integrated into the criteria for patient subgroups, such as age, sex, and lifestyle choices. Observation based on age strata showed significant differences in cancer biology and immune functions in older vs. younger patients [173]. The published reports illustrated that ICIs are more effective in the elderly, which may be due to the upregulated expression of immune checkpoints with age [174]. However, the physiological changes in the elderly may influence the pharmacology of anticancer drugs. As an example, in patients older than 75 years, CDK4/6 inhibitors induced higher rates of adverse effects, which decreased the quality of life [175]. These studies informed clinicians about making tailored therapeutic strategies for each patient subgroup, maximizing efficacy and minimizing toxicity.

Despite ICIs, novel immunotherapy approaches such as cellular therapy and cancer vaccine are emerging. Adoptive cellular therapy (ACT) has been approved to treat hematologic malignancies, but its exploitation is hindered in solid tumors. One of the barriers is immunosuppressive TIME. Applying targeted agents to directly eliminate or reprogram immunosuppressive cells is a promising approach to improving the efficacy of ACT. For instance, CDK4/6 inhibitors reduce the proportion of MDSC and Treg cells in patients with metastatic breast cancer. Therefore, using targeted agents simultaneously with or before ACT may enhance the antitumor response.

Reusing molecularly targeted agents as an adjuvant in immunotherapy is a field that merits further study. Taken together, targeted therapy in combination with immunotherapies has shown great potential, which can lead to superior clinical efficacy.

## Figures and Tables

**Figure 1 pharmaceutics-14-01768-f001:**
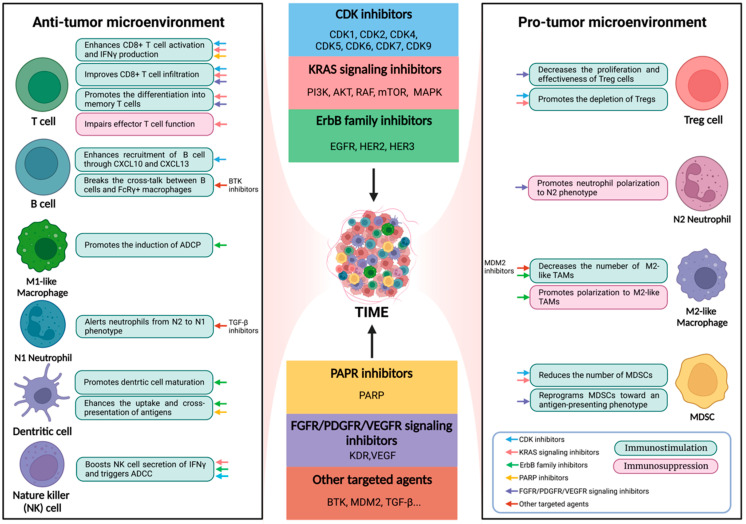
**Modulatory Effects of Targeted Therapy on Immune Cells.** The molecularly targeted agents function as immunostimulatory (the green block diagram) or immunosuppressive (the red block diagram) modulators in TIME. The color of each little arrow matches the corresponding drug category in the middle of figure. ADCC, antibody-dependent cell cytotoxicity; ADCP, antibody-dependent cellular phagocytosis; AKT, V-akt murine thymoma viral oncogene homolog; BTK, Bruton’s tyrosine kinase; CDK, cycle-dependent kinase; EGFR, epidermal growth factor receptor; FGFR, fibroblast growth factor receptor; HER, human epidermal growth factor receptor; KDR, kinase insert domain receptor; MAPK, mitogen-activated protein kinases; MDM2, mouse double minute 2; MDSC, myeloid-derived suppressor cell; mTOR, mechanistic target of rapamycin; PARP, poly(ADP-ribose) polymerase; PDGFR, platelet-derived growth factors receptor; PI3K, phosphatidylinositol-3-kinase; TAM, tumor-associated macrophage; TIME, tumor immune microenvironment; TGF-β, transforming growth factor-β; Treg, regulatory T cell; VEGFR, vascular endothelial growth factor receptor (created with BioRender.com (accessed on 11 August 2022)).

**Table 1 pharmaceutics-14-01768-t001:** Clinical trials of ICIs plus targeted agents targeting BRAF/MEK, CDK, or ErbB.

Cancer Type	Targeted Agents	Combination Agents	Phase	Control	Primary Endpoint	NCT Number
BRAF/MEK Inhibitors						
Melanoma *	Cobimetinib + Vemurafenib	Atezolizumab	III	Cobimetinib + Vemurafenib +	PFS (vs. control): 15.1 vs. 10.6 months, HR = 0.78	NCT02908672
				placebo		
Melanoma	Trametinib + Dabrafenib	Ipilimumab + Nivolumab	III	Arm A: N/I + D/T	2-year OS rate (Arm A vs. Arm B): 72% vs. 52%	NCT02224781
				Arm B: D/T + N/I		
CDK Inhibitors						
Breast Cancer	Palbociclib	Pembrolizumab + Letrozole	I/II	-	ORR: 56%	NCT02778685
EGFR Inhibitors						
HNSCC	Cetuximab	Pembrolizumab	II	-	ORR: 45% (95% CI 28–62)	NCT03082534
CRC	Panitumumab	Ipilimumab + Nivolumab	II	-	ORR: 35% (95% CI 21–48)	NCT03442569
CRC	Cetuximab	Pembrolizumab	Ib/II	-	A manageable safety profile	NCT02713373
HER2						
HNSCC	Afatinib	Pembrolizumab	II	-	High PD-L1 expression (TPS ≥ 50 ORR: 71%, CPS ≥ 20 ORR: 63%);	NCT03695510
					EGFR amplification (ORR: 100%);	
					MTAP loss or mutation (ORR: 0%)	
Oesophageal, G/GEJ Cancer	Trastuzumab	Pembrolizumab	II	-	6-months PFSR: 70% (95% CI 54–83)	NCT02954536
Breast Cancer	Trastuzumab	Pembrolizumab	Ib/II	-	ORR: 15% (90% CI 7–29)	NCT02129556

*: Approved by FDA. Source: http://www.clinicaltrials.gov (accessed on 29 July 2022). CRC, colorectal cancer; CDK, cycle-dependent kinase; CI, confidence interval; CPS, combined positive score; EGFR, epidermal growth factor receptor; HNSCC, head and neck squamous cell carcinoma; HR, hazard ratio; MEK, mitogen-activated protein kinase; MTAP, methylthioadenosine phosphorylase; ORR, overall response rate; PFS, progression-free survival; PFSR, progression-free survival rate; TPS, tumor proportion score.

**Table 2 pharmaceutics-14-01768-t002:** Clinical trials of ICIs plus targeted agents targeting FGFR/PDGFR/VEGFR.

Cancer Type	Targeted Agents	Combination Agents	Phase	Control	Primary Endpoint	NCT Number
Endometrial Cancer *	Lenvatinib	Pembrolizumab	III	Chemotherapy	Median OS (vs. control): 17.4 vs. 12.0 months, HR = 0.68	NCT03517449
					Median PFS (vs. control): 6.6 vs. 3.8 months, HR = 0.60	
HCC *	Bevacizumab	Atezolizumab	III	Sorafenib	Median OS (vs. control): 19.2 vs. 13.4 months, HR = 0.66	NCT03434379
					Median PFS (vs. control): 6.9 vs. 4.3 months, HR = 0.65	
RCC *	Lenvatinib	Pembrolizumab	III	Lenvatinib +	PFS (vs. control): 23.9 vs. 9.2 months, HR = 0.39	NCT02811861
				Sunitinib		
RCC *	Axitinib	Pembrolizumab	III	Sunitinib	OS (vs. control): median not reached vs. 35·7 months, HR = 0.68	NCT02853331
RCC *	Cabozantinib	Nivolumab	III	Sunitinib	PFS (vs. control): 16.6 vs. 8.3 months, HR = 0.51	NCT03141177
RCC	Bevacizumab	Atezolizumab	III	Sunitinib	PD-L1 positive group median PFS (vs. control): 11.2 vs. 7.7 months, HR = 0.74	NCT02420821
Breast Cancer	Apatinib	Camrelizumab	II	-	ORR: 43.3% (95% CI 25.5–62.6)	NCT03394287
CRC	Regorafenib	Avelumab	II	-	CR, PR: 0; SD: 57.5%	NCT03475953
DTC	Lenvatinib	Pembrolizumab	II	-	ORR: 62%	NCT02973997
Endometrial Cancer	Lenvatinib	Pembrolizumab	II	-	ORR: 38% (95% CI 28.8–47.8)	NCT02501096
GC	Lenvatinib	Pembrolizumab	II	-	ORR: 69% (95% CI 49–85)	NCT03609359
Ovarian Cancer	Bevacizumab	Nivolumab	II	-	ORR: 21%	NCT02873962
CRC	Regorafenib	Toripalimab	Ib/II	-	ORR: 15.2% (95% CI 5.7–32.7)	NCT03946917
Pancancer	Pemigatinib	Pembrolizumab	I/II	-	A manageable safety profile and pharmacodynamic and clinical activity	NCT02393248
Sarcomas	Sunitinib	Nivolumab	Ib/II	-	6-months PFSR: 48% (95% CI 41–55)	NCT03277924
Gastric Cancer/CRC	Regorafenib	Nivolumab	Ib	-	Safety: Regorafenib 80 mg plus nivolumab	NCT03406871
HCC	Lenvatinib	Pembrolizumab	Ib	-	Promising antitumor activity with a tolerable safety profile	NCT03006926
NSCLC	Anlotinib	Sintilimab	Ib	-	ORR: 72.7% (95% CI 49.8–89.3)	NCT03628521
HCC	Apatinib	Camrelizumab	I	-	ORR: 30.8% (95% CI 17–47.6)	NCT02942329
NSCLC, G/GEJ Cancer, UC	Ramucirumab	Pembrolizumab	I	-	A manageable safety profile with favorable antitumor activity	NCT02443324

*: Approved by FDA. Source: http://www.clinicaltrials.gov (accessed on 29 July 2022). CI, confidence interval; CR, complete remission; CRC, colorectal cancer; DTC, differentiated thyroid cancer; GC, gastric cancer; HCC, hepatocellular carcinoma; HR, hazard ratio; NSCLC, non-small cell lung cancer; ORR, overall response rate; OS, overall survival; PFS, progression-free survival; PFSR, progression-free survival rate; PR, partial remission; RCC, renal cell carcinoma; UC, urothelial carcinoma.

**Table 3 pharmaceutics-14-01768-t003:** Clinical trials of ICIs plus epigenetic agents.

Cancer Type	Targeted Agents	Combination Agents	Phase	Control	Primary Endpoint	NCT Number
Uveal Melanoma	Entinostat	Pembrolizumab	II	-	ORR: 14% (95% CI 3.9–31.7)	NCT02697630
AML	Azacitidine	Nivolumab	II	-	ORR: 33%	NCT02397720
Hodgkin Lymphoma	Decitabine	Camrelizumab	II	-	ORR: 60% (95% CI 45–74)	NCT02961101

Source: http://www.clinicaltrials.gov (accessed on 29 July 2022). AML, acute myeloid leukemia; CI, confidence interval; ORR, objective response rate.

## Data Availability

All data supporting reported results can be found in this manuscript.

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
