# Peer review of "Immune Modulatory Effects of Molecularly Targeted Therapy and Its Repurposed Usage in Cancer Immunotherapy"

_pharmaceutics, 2022, doi:10.3390/pharmaceutics14091768_

Round 1
Reviewer 1 Report
The review by Zhang et al is an interesting compendium of potential combinatorial therapies between small molecules and more common immunotherapeutic strategies in cancer. In particular, they focus on how it affects specific cells in the tumor microenvironment. The main concerns are 1) the need for English editing as there are several grammatical mistakes interspersed in the manuscript. 2) The introduction makes it seem that it is about mutational targeting which is not.
My suggestions are these:
1) Language editing.
2) Figure 1 needs to be clarified. I think they are trying to convey that targeting specific mutations affects specific cells? Still, it is confusing and not very informative as is. Perhaps doing instead several figures on the TIME and focusing on specific drugs?
3) Adding figures to aid the reader capture the effects on specific cell lineages in TIME.
Reviewer 2 Report
The review requires editing. The introduction and figure 1 have to be modified. The figure is confusing, It lacks a proper legend, and it is not correct since not all cells on the lest of the figure are stimulated equally. There should be a figure on the pharmacological targets. The description of the cells is vague, and it does provide a clear insight into the next section on treatment. The section on clinical trials can also be improved based on the success rate. Finally, the abstracts and conclusions are too short and should be changed
Reviewer 3 Report
Zhang and colleagues presented a review article aimed at describing the role of targeted therapy in immune stimulation or inhibition and the consequent effects of such treatments on the efficacy of immune checkpoint inhibitors. Overall, the authors described different relevant drugs currently adopted for the treatment of several tumors. However, the description of targeted therapies and their effects on the immune system is not systematic but narrative. In my opinion, this represents a limitation of the study that the authors have to claim in the Conclusive remarks. Overall, the manuscript is well written, however, some information is missing and references should be added/updated. Below are reported minor/major comments that will improve the quality of the manuscript:
1) Please add a legend at the bottom of Figure 1. In particular, what are the green spheres within vessels and tumor bulk? Same comment for the blue and yellow cells within the tumor;
2) In the first part of the Introduction, the authors should briefly introduce the evolution of cancer pharmacological treatments describing the fast development of novel and effective therapeutic approaches in cancer;
3) Please provide supporting references for the following sentences: “Targeted agents have become the first- or second-line treatments for most advanced malignancies, including breast cancer, lung cancer, colorectal carcinoma (CRC), hepatocellular carcinoma (HCC), renal cell carcinoma (RCC) and others. Targeted agents exert broad anti-tumor activity via direct inhibition on tumor cells and indirect impacts on the tumor immune microenvironment.”. For this purpose, please see:
- PMID: 30483135
- PMID: 34408877
- PMID: 31425592
- PMID: 31807576
4) Provide references for the following sentence: “Tumor cells can produce immune suppressive factors, such as IL-10, IL-8, and transforming growth factor-β (TGF-β), and reprogram the metabolic process to alter the TIME, which favors tumor progression.”. Please see:
- PMID: 27348007
- PMID: 34576335
- PMID: 30324115
- PMID: 33804410
5) The following sentence is wrong: “A few cancer types, such as pancreatic ductal adenocarcinoma (PDAC), neuroendocrine neoplasm, and mismatch repair-proficient CRC, barely benefit from immunotherapy, which may be due to the infiltration of immunosuppressive cells, exhaustion of effector T cells, or “cold” immune phenotype[6,11].”. Several studies demonstrated the efficacy of ICIs in different other tumors including breast cancer, melanoma and NSCLC where T cell immune infiltrates (TILs) play an active role in cancer progression and response to therapy. Please see:
- PMID: 34944986
- PMID: 34853355
- PMID: 30221052
6) In Chapter 4.2 the authors have to better emphasize the crosstalk existing between the RAS and PI3K pathway and the strategies to overcome drug resistance related to activating mutation of PIK3CA, BRAF, or other genes of the pathway. For this purpose, please see:
- PMID: 35335966
- PMID: 35806358
- PMID: 21829508
- PMID: 33081092
- PMID: 32605090
7) Throughout the manuscript there are some minor grammar errors. Please check the entire manuscript and correct the errors.
Round 2
Reviewer 1 Report
They have addressed all my comments. I am particularly impressed by the new Figure 1 which is now really informative and helpful.
Reviewer 2 Report
The manuscript was improved partially. Figure 1 is still complex. The top of the figure illustrates better the text; however, the colours used can be enhanced. The tables were improved, and recent data was incorporated. The discussion needs to be focused on new propositions, new targets, or combined therapies, ie. the first-line therapy in old vs young individuals, the use of tyrosine kinase inhibitors with checkpoint inhibitors for older patients as compared with cytotoxic therapies for young patients. The idea is to decrease the tumour burden and increase survival without reducing the quality of life.
Reviewer 3 Report
The authors well addressed almost all of my previous comments. As a further minor comment, in Tables 1, 2, 3 please add if the treatments described are FDA or EMA approved (consider to add an additional column if necessary).
